# Effects of Deep Vertical Rotary Tillage Management Methods on Soil Quality in Saline Cotton Fields in Southern Xinjiang

Zhijie Li [1,2], Hongguang Liu [1,2,*], Haichang Yang [3] and Tangang Wang [4,*]

1   College of Water Conservancy & Architectural Engineering, Shihezi University, Shihezi 832000, China; lzhijie_2023163@163.com
2   Xinjiang Production & Construction Group Key Laboratory of Modern Water-Saving Irrigation, Shihezi 832000, China
3   College of Agronomy, Shihezi University, Shihezi 832000, China; yhc2012@126.com
4   Institute of Agricultural Science of the Third Division of Xinjiang Production and Construction Corps, Tumushuk 843900, China
*   Correspondence: liuhongguang-521@163.com (H.L.); ssnkswtg@163.com (T.W.)

**Abstract:** A long-term high-saline soil environment will limit the improvement of soil quality and cotton yield. Modified tillage management measures can improve soil quality, and the establishment of a soil quality evaluation system will facilitate evaluation of the soil quality and land production potential in southern Xinjiang. The objective of this study was to determine the effects of different tillage management methods on soil quality in saline cotton fields in southern Xinjiang. A three-year experiment was conducted in Tumushuke, Xinjiang, with different deep vertical rotary tillage depths (DTM20, 20 cm; DTM40, 40 cm; DTM60, 60 cm) and conventional tillage (CTM, 20 cm). The soil quality index (SQI) under different tillage management methods was established by using the full dataset (TDS) with a scoring function for eight indicators, including physicochemical properties of the soil from 0 to 60 cm, to evaluate its impact on the soil quality of the saline farmland in southern Xinjiang. The results of the study showed that deep vertical rotary tillage management can effectively optimize soil structure; reduce soil bulk density (BD), soil solution conductivity (EC), and pH; and promote the accumulation of soil organic carbon (SOC) and total nitrogen (TN) in the soil. However, the average diameter of soil water-stable aggregates (MWD) in a 0–60 cm layer becomes smaller with an increasing depth of tillage. This does not reduce crop yields but does promote soil saline leaching. In addition, the significant linear relationship ($p < 0.001$) between seed cotton yield and soil quality indicated that improving soil quality was favorable for crop yield. The principal component analysis revealed BD, MWD, pH, and EC as limiting sensitive indicators for seed cotton yield, while SOC and TN were positive sensitive indicators. The soil quality index (SQI) values of DT40 and DTM60 were significantly higher than that of CTM by 11.02% and 15.27%, respectively. Overall, the results show that DTM60 is the most suitable tillage strategy to improve soil quality and seed cotton yield in this area, and this approach will provide a reliable theoretical basis for the improvement of saline farmland.

**Keywords:** soil quality; saline farmland; deep vertical rotary tillage; soil tillage management methods; seed cotton yield





## 1. Introduction

Xinjiang in northwestern China is the largest cotton-producing region in the country, with an annual output that can reach 1.8 million t, accounting for 93.6% of China's total production [1]. Cotton production areas located in the southern part of Xinjiang have long been constrained by drought and water scarcity with soil salinization [2]. At the same time, under the long-term cotton cropping and shallow tillage management system based on five-share plowing, soil salinity has been accumulating in the root zone, limiting the improvement of cotton yield [3]. Increasing soil salinity problems and global climate

change can lead to degradation of cropland, and rational tillage practices are an effective way to improve soil quality and crop yield potential [4].

Poor soil structure and low soil fertility caused by soil salinization, and high salinity in the soil profile are key factors limiting cotton production in southern Xinjiang [5]. Saline soils can be effectively improved through modified soil tillage management methods to increase soil fertility and improve soil quality [6]. Traditionally, once or twice a year, plowing (spring or autumn plowing) is used in the Xinjiang region to form a loose soil structure, and spring and winter irrigation is used to wash soil saline ions [7]. However, long-term shallow tillage increases soil compaction and shallow tillage layer, which severely limits soil water vapor transmission [8]. Deep tillage is a typical conservation tillage practice aimed at conservation that can achieve soil loosening without disturbing the soil layers at all depths [9]. The improved soil structure and loosened subsoil allow a crop root system to obtain more water and nutrient resources for higher overall crop yields [10–12]. In order to explore the improved soil and tillage management in southern Xinjiang, this work evaluated the use of deep vertical rotary tillage technology. This method can realize different depths of deep tillage by controlling the lifting and lowering of the auger head and rapid rotary grinding and crushing of the soil, which suspends into a ridge and does not disturb the upper soil layer [13]. This tillage method has been widely promoted in China, and it can better harmonize soil water, air, and heat to build a healthy soil tillage layer. Given the water-scarce climate and soil salinization at the southern border, the type of tillage management is closely related to the agro-environmental conditions [14]. The goal of this work was to evaluate the ability of this tillage method to improve soil quality and sustainable agricultural development.

The soil quality index (SQI) is an assessment tool that includes physical, chemical, and biological indicators, and can be used to estimate changes in soil conditions in arable soils over time due to land-use practices and soil management [15]. For effective evaluation, soil quality indicators should be physical, chemical, and biological properties that are sensitive to soil management methods; combining these soil properties into a single indicator can make the assessment more meaningful and practical. But soil quality sensitivity indicators vary considerably from region to region [16]. Thus, for meaningful evaluation of tillage management in southern Xinjiang, we needed to first determine the sensitive indicators affecting soil quality in this region.

Saline soil in Xinjiang is an important reserve arable land resource in China, but agricultural production in this region is limited by water scarcity and soil salinization [17]. Thus, there is significant interest in developing new strategies for improved utilization of saline farmland resources for more sustainable agricultural development. In this study, three different types of deep vertical rotary tillage and one type of conventional tillage were compared for tillage of salinized farmland in southern Xinjiang from 2020 to 2022. By measuring its soil physicochemical properties and yield, analyzing the effects of different tillage management methods on it, and obtaining the soil quality index (SQI) and sensitivity indices under different tillage management methods, the overall functional capacity of the soil in the region was finally determined. The effects of the different tillage management methods on soil physicochemical properties and yield were determined by obtaining the soil quality index (SQI) values to assess the overall functional capacity of the soil in the region. The results of this work should provide a theoretical basis to improve salinized farmland in the southern Xinjiang region.

## 2. Materials and Methods

Experiments were conducted in a heavily salinized agricultural field in Tumushuke, Xinjiang, China (79°2′5″ E, 40°0′10″ N; altitude 1098 m). The study area has a temperate extremely arid desert climate, with an average annual precipitation of 38.3 mm, an annual evapotranspiration of 1643 to 2202 mm, and an average annual temperature of 11.6 °C. The total precipitation during the experimental period of 2020–2022 was 264 mm, and the average air temperature was 21.61 °C. The total precipitation during the test period was

264 mm, and the average air temperature was 21.61 °C (June 2020–September 2022). The topsoil of the region has a clayey texture and is heavily salinized, with a salt composition dominated by chlorides [18], but there is plenty of sunshine, and cotton is the main cash crop grown in the area. The groundwater table is 7.2–8 m in the irrigated season and 8–10 m in the non-irrigated season. The soil basic information was measured at 0–60 cm in the test area, and the results are shown in Table 1.

**Table 1.** Soil texture of the soil in the study area.

| Soil Depth cm | Soil Fraction/% | | | Soil Texture [1] |
| | Sand Particles 0.05–2/mm | Silt Particles 0.05–0.002/mm | Clay Particles <0.002/mm | |
| --- | --- | --- | --- | --- |
| 0–20 | 20.80 | 70.77 | 8.43 | Silty loam |
| 20–40 | 14.56 | 76.38 | 9.06 | Silty loam |
| 40–60 | 9.97 | 80.49 | 9.54 | Silty loam |

[1] Soil particles were graded according to the USDA Soil Taxonomy system of Soil Classification Standards [19].

### 2.1. Experimental Site and Experimental Design

Twelve experimental plots were established, each with an area of $14 \times 15$ m$^2$. Four treatments were tested, and each treatment was replicated three times, with a 10 m isolation zone between neighboring plots to ensure that the treatments were not affected by nearby plots. The treatments were categorized into different deep vertical rotary tillage depths (DTM20, 20 cm; DTM40, 40 cm; DTM60, 60 cm) and conventional tillage (CTM, 20 cm). The deep vertical rotary tillage treatments were started in April 2020 using a loosening machine (Aksu Wufeng Agricultural Machinery Co., Ltd., Aksu, China, Model 1FSGL-230) to achieve different tillage depths, and in 2021 and 2022, a conventional five-share plow was used (Anhui Huaifeng Modern Agricultural Equipment Co., Ltd., Hefei, China, Model 1L-530J). Conventional tillage treatment (CTM) using a five-share plow for tillage in 2020–2022 (Table 2) and spring irrigation (2800 m$^3 \cdot$ha$^{-1}$) was performed after completion of tillage. The test variety of "Xinluzhong No. 56" cotton was sown in the spring after irrigation, and the straw was crushed and returned to the field after harvesting in September, with a deep turning of 30 cm. Drip irrigation under the membrane was adopted, with the drip irrigation tapes arranged in three tubes and six rows (Figure 1). The drip irrigation tapes had a diameter of 16 mm, a wall thickness of 0.2 mm, a drip head spacing of 30 cm, and a flow rate of 3.2 L$\cdot$h$^{-1}$. Irrigation water was mixed with water from wells and canals, with a ratio of irrigation of 1:1. The irrigation and fertilization system was consistent for each year during the experimental period, and the cotton was irrigated 10 times during the season, with an interval of 7–10 d between each irrigation treatment. The total irrigation quota was 47,400–48,000 m$^3 \cdot$ha$^{-1}$. Water-soluble fertilizers were applied with each irrigation treatment, including 500 kg$\cdot$ha$^{-1}$ of urea (N mass fraction $\geq$ 46%), 40 kg$\cdot$ha$^{-1}$ of potassium xanthate (mass fraction of xanthate $\geq$ 50%), 290 kg$\cdot$ha$^{-1}$ of high-nitrogen and high-phosphorus water-soluble fertilizers (N + P$_2$O$_5$ mass fraction $\geq$ 74%), and 210 kg$\cdot$ha$^{-1}$ of high-nitrogen and high-potassium water-soluble fertilizers (N + K$_2$O mass fraction $\geq$ 70%). Other field management measures were performed according to local implementation.

**Table 2.** Soil tillage management methods for each treatment in the experimental area.

| Soil Tillage Management Methods | Tillage Machinery and Depth | |
| | 2020 | 2021–2022 |
| --- | --- | --- |
| CTM | Five-share plow machinery, 20 cm | Five-share plow machinery, 20 cm |
| DTM20 | Deep vertical rotary tillage, 20 cm | Five-share plow machinery, 20 cm |
| DTM40 | Deep vertical rotary tillage, 40 cm | Five-share plow machinery, 20 cm |
| DTM60 | Deep vertical rotary tillage, 60 cm | Five-share plow machinery, 20 cm |

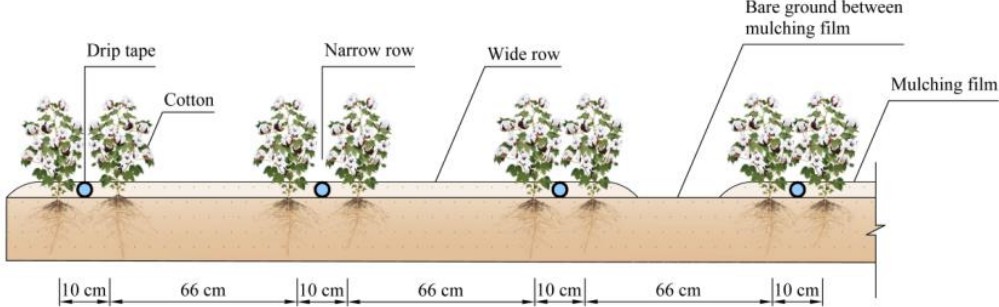

**Figure 1.** Schematic diagram of cotton planting.

### 2.2. Collection and Analysis of Soil Samples

On the third day after each irrigation, samples were randomly taken from the wide rows of each treatment to a depth of 60 cm using the "five-point method", and the average value during the reproductive period was taken as the average value of the treatment at a depth of 0–60 cm. A portion of the collected samples was used for pH measurement with a pH meter (Shanghai Yidian Scientific Instrument Co., Ltd., Shanghai, China, Model PHS-3C) and soil EC measurement by a conductivity meter (Shanghai Yidian Scientific Instrument Co., Ltd., Shanghai, China, Model DDS-307). The rest of the soil was used for the determination of soil nutrients, in which soil organic carbon was determined by the potassium dichromate-sulfuric acid external heating method (SOC), total nitrogen was determined by Kjeldahl nitrogen fixation (TN), total potassium was determined by $H_2SO_4$-$H_2O_2$ boiling and the flame photometric method (TK), and total phosphorus was determined by $H_2SO_4$-$H_2O_2$ boiling and the molybdenum blue method with ascorbic acid (TP) [20].

Cotton harvesting was performed in September each year, and the soil samples were collected and air-dried using the "five-point method." For each soil sample, 100 g of soil were weighed after thorough mixing, and the soil water stability aggregates were determined using the wet sieve method [21]. The average weight diameter (MWD) of the soil water stability aggregates from 0 to 60 cm was calculated by the following formula:

$$\text{MWD} = \frac{\sum_{i=1}^{n} d_i w_i}{\sum_{i=1}^{n} w_i} \tag{1}$$

where $w_i$ is the proportion of agglomerate mass in each particle size range, %; $d_i$ is the average diameter of agglomerates in any level range, mm.

Soil samples were collected using the cutting-ring method between the wide rows of each treatment to a depth of 60 cm during the cotton harvest in September of each year, and each treatment was replicated three times. The soil bulk density (BD) was calculated using the following formula, which was calculated to obtain the mean value of 0–60 cm:

$$d_v = \frac{M \cdot 100}{V(100 + W)} \tag{2}$$

where, dv is the soil bulk density, g·cm$^{-3}$; M is the weight of wet soil, g; V is the volume of the ring knife, cm$^3$; and W is the soil moisture content in the ring knife, %.

Seed cotton yield was obtained by hand-picking, drying, and weighing 100 randomly selected bolls during the cotton harvest in September each year. Individual boll weight was calculated by dividing the total weight of 100 bolls by the total number of bolls, and the final average was taken and substituted into the following equation as seed cotton yield for each treatment. Each treatment was repeated three times.

$$Y = y \times a \times m \tag{3}$$

where, Y is the seed cotton yield, kg·ha$^{-1}$; y is the boll weight, g; a is the number of bolls, per plant; m is the planting density, plant·ha$^{-1}$.

### 2.3. Soil Quality Assessment

Physical and chemical indicators commonly used in SQI evaluation were initially selected based on previous studies. The selected indicators were used to establish a total dataset (TDS) and included soil physical property indicators (BD, MWD) [22] that can reflect the effects of different tillage management practices on soil structure and soil particle distribution, and soil chemical property indicators (EC, pH, SOC, TN, TP, TK) [22] that can reflect the effects of different tillage management practices on soil ecology and fertility [23]. The soil indicators in TDS were transformed into normalized values between 0.1 and 1.0 using three types of scoring functions. The positive S-type (SSF$_1$) function was applied to positive slopes, the inverse S-type (SSF$_2$) function was applied to negative slopes, and the parabolic (SSF$_3$) function was applied to positive slopes that change to negative slopes at a certain threshold [15]. The standard scoring functions used for normalization of soil quality indicators and their thresholds are shown in Table 3. The score curve equation was used to calculate the soil indicator scores, and the SSF formula shown in Table 3 is as follows [24].

$$\mathrm{TypeS(SFF_1):\ f(x)} = \begin{cases} 1.0 & (\mathrm{x} \geq \mathrm{b}) \\ \frac{\mathrm{x}-\mathrm{a}}{\mathrm{b}-\mathrm{a}} & (\mathrm{a} < \mathrm{x} < \mathrm{b}) \\ 0.1 & (\mathrm{x} \leq \mathrm{a}) \end{cases} \tag{4}$$

$$\mathrm{TypeS(SFF_2):\ f(x)} = \begin{cases} 0.1 & (x \geq b) \\ \frac{\mathrm{x}-\mathrm{a}}{\mathrm{b}-\mathrm{a}} & (\mathrm{a} < \mathrm{x} < \mathrm{b}) \\ 1.0 & (\mathrm{x} \leq \mathrm{a}) \end{cases} \tag{5}$$

$$\mathrm{TypeS(SFF_3):\ f(x)} = \begin{cases} 0.1 & (\mathrm{x} \leq \mathrm{a},\ \mathrm{x} \geq \mathrm{b}) \\ \frac{\mathrm{x}-\mathrm{b}}{\mathrm{b_2}-\mathrm{b}} & (\mathrm{b_2} < \mathrm{x} < \mathrm{b}) \\ \frac{\mathrm{x}-\mathrm{a}}{\mathrm{b_1}-\mathrm{a}} & (\mathrm{a} < \mathrm{x} < \mathrm{b_1}) \\ 1.0 & (\mathrm{b_1} < \mathrm{x} < \mathrm{b_2}) \end{cases} \tag{6}$$

where f(x) is the soil quality indicator score; x is the measured value of the soil indicator; and a, b, b$_1$, and b$_2$ are the critical values, see Table 3.

**Table 3.** Threshold values and standardized scoring functions used for soil quality indicators.

| Factor | Unit | Scoring Survey | a * | b * | b$_1$ * | O * | b$_2$ * | Reference Source |
|---|---|---|---|---|---|---|---|---|
| BD | g·cm$^{-3}$ | SSF$_2$ | 1.3 | 1.8 | | | | [25] |
| MWD | mm | SSF$_1$ | 0.4 | 2.0 | | | | [26] |
| EC | mS·cm$^{-1}$ | SSF$_2$ | 0.2 | 4 | | | | [25] |
| pH | | SSF$_3$ | 3 | 11 | 5.5 | 7 | 8.5 | [25] |
| SOC* | g·kg$^{-1}$ | SSF$_1$ | 3.48 | 23.2 | | | | |
| TN | g·kg$^{-1}$ | SSF$_1$ | 0.5 | 20 | | | | [18] |
| TP | g·kg$^{-1}$ | SSF$_1$ | 0.2 | 1 | | | | |
| TK | g·kg$^{-1}$ | SSF$_1$ | 5 | 25 | | | | |

* a = lower threshold at which or below the score is 0.1; b = upper threshold at which or above score is 1.0; b$_1$ = lower baseline, at which score is 0.5 with bell-shaped relationship; O = optimum level, at which score is 1.0 with bell-shaped relationship; b$_2$ = upper baseline at which score is 0.5 with bell-shaped relationship. The threshold for soil organic carbon was obtained by conversion based on organic matter = organic carbon × 1.742 [25].

The weights of the indicators can be determined by dividing the common factor of principal component analysis by the total eigenvalues. Finally, after scoring and weighting the selected indicators, the SQI was calculated using the soil quality index formula:

$$\mathrm{SQI} = \sum_{i=1}^{n} \mathrm{W_i} \times \mathrm{S_i} \tag{7}$$

where $W_i$ is the indicator weight value, $S_i$ is the indicator score, and n is the number of variables integrated in the indicator.

### 2.4. Data Processing

Experimental data processing was performed using Excel 2019. SPSS Statistics 22.0 (SPSS Inc., Chicago, IL, USA) was used for normal distribution tests, factor analysis, and Pearson correlation analysis. After testing, the data were normally distributed; MANOVA was used to test the differences in soil quality indicators between tillage years and tillage management practices. Origin 2021 was used to plot the principal component analysis to clarify the relationship between soil quality indicators and yield.

## 3. Results

### 3.1. Multivariate Analysis of Variance (MANOVA)

The eight selected soil quality indicators were significantly affected by period of tillage, tillage management practices, and their interaction effects (Table 4). We analyzed the data for different soil tillage practices in each period and investigated the effect of soil tillage methods on soil quality indicators in 0–60 cm and seed cotton yield.

**Table 4.** Multivariate analysis of variance (MANOVA) results to assess the effect of period, treatment, and their interactions for eight measured soil properties and seed cotton yield.

| Factors * | DF | Wilk's λ | *p*-Value |
|---|---|---|---|
| Period of tillage | 18 | 0.004 | <0.001 |
| Tillage management | 27 | 0.001 | <0.001 |
| Period of tillage × Tillage management | 54 | 0.017 | 0.002 |

* The results of the data normality tests are detailed in Table S3.

### 3.2. Soil Properties

TP and TK in the 0–60 cm soil depth range during 2020–2022 were not significantly different between groups among treatments, but there were significant differences between groups of soil indicators including BD, MWD, pH, EC, SOC, and TN (Figure 2). The mean values of BD, MWD, pH, and EC under different tillage management at three years were in the order of CTM > DTM20 > DTM40 > DTM60, while the opposite pattern was seen for SOC and TN. Deep vertical rotary tillage management consistently affected the quality indicators of 0–60 cm soils during the three years (2020–2022). DTM significantly reduced BD and pH compared to CTM, but there was no significant difference in pH between DTM20 and CTM. MWD and EC were significantly lower than those of CTM by 3.79%, 6.40%, and 29.40%, 38.21% for DTM40 and DTM60, respectively; the difference between CTM and DTM20 was not significant. SOC of DTM20, DTM40, and DTM60 were significantly higher than CTM by 2.32%, 4.51%, and 7.47%, respectively; and TN of DTM40 and DTM60 were significantly higher than CTM by 1.51% and 2.97% (the difference between CTM and DTM20 was not significant). There were no significant differences in TP and TK between deep vertical rotary tillage and conventional tillage ($p > 0.05$). Neither deep vertical rotary tillage management nor an increase in the number of years of tillage significantly affected the content of TP and TK in the soil.

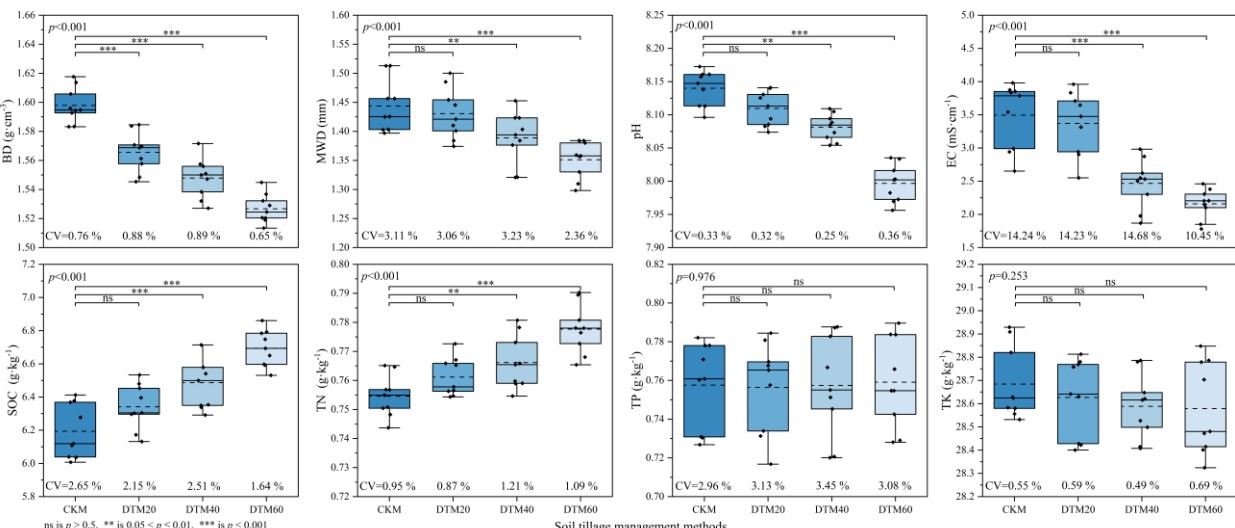

**Figure 2.** Boxplot and coefficient of variation of eight soil properties of the 0–60 cm layer in 2020, 2021, and 2022. CV is a variable coefficient of each soil property in 2020, 2021, and 2022. *p* is the level of significance of soil properties between treatments in 2020, 2021, and 2022. BD, soil bulk density; MWD, soil mean weight diameter; EC electrical conductivity; SOC, soil organic carbon; TN, total nitrogen; TK, total potassium; TP, total phosphorus. CTM, conventional tillage; DTM20, 20 cm; DTM40, 40 cm; DTM60, 60 cm, the same as below. Where ns is $p > 0.05$, ** is $0.05 < p \leq 0.01$, *** is $p < 0.001$.

MWD showed a significant positive correlation with BD (0.51), EC showed a significant positive correlation with BD (0.80), and pH showed a significant positive correlation with BD, MWD, and EC (0.54, 0.57, and 0.74, respectively). SOC showed a significant negative correlation with BD, MWD, EC, and pH (−0.83, −0.37, −0.90, and −0.70, respectively). TN showed a significant negative correlation with BD, MWD, EC, and pH (−0.66, −0.49, −0.38, and −0.41, respectively). TN showed a significant positive correlation with SOC (0.46). MWD and pH were significantly positively correlated with TK (0.44 and 0.74, respectively). TP showed no correlation with any of the indicators (Figure 3).

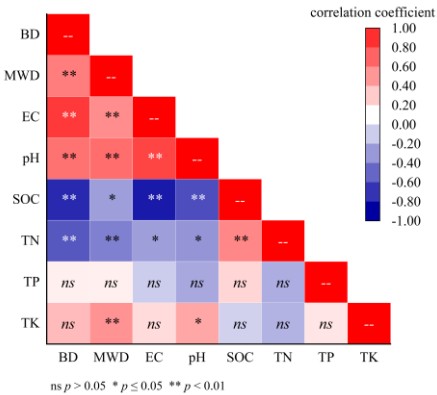

**Figure 3.** Correlation coefficients among soil properties in the 0–60 cm layer. The correlation coefficients and levels between soil properties are shown in the figure. Where ns is $p > 0.05$, * is $p \leq 0.05$, ** is $p < 0.001$.

### 3.3. Soil Quality

The common factor variances of the eight soil property indicators were obtained by factor analysis, and all were >0.5 (Table 5). Three components with eigenvalues greater than one were selected, and these components explained a total of 82.53% of the variance in the dataset. The eigenvalues of PC1, PC2, and PC3 were categorized as 3.51, 1.72, and 1.38, explaining 43.88%, 21.45%, and 17.20% of the variance in the dataset, respectively. Both

PC1 and PC2 consisted of positive principal component coefficients, or loadings, on BD, MWD, TK, and loadings on TN. SOC and TP were loadings in PC1 and positive loadings in PC2. Indicators of pH and EC were positive loads in PC1 and loads in PC2. MWD, pH, SOC, TN, and TK were positive loadings in PC3, and BD, EC, and TP were loadings. In the TDS dataset, SOC had the largest common factor variance and weight, 0.91 and 0.14, respectively, and MWD had the smallest, 0.68 and 0.10.

**Table 5.** Soil properties of factor pattern, common factor variance, and weighting considered in TDS.

| Soil Properties | Packet | PCA | | | Communality | Weighting |
|---|---|---|---|---|---|---|
| | | PC1 | PC2 | PC3 | | |
| BD | 1 | 0.886 | 0.083 | −0.288 | 0.875 | 0.1325 |
| MWD | 1 | 0.687 | 0.297 | 0.350 | 0.683 | 0.1035 |
| pH | 1 | 0.826 | −0.262 | 0.279 | 0.828 | 0.1254 |
| EC | 1 | 0.875 | −0.312 | −0.179 | 0.895 | 0.1356 |
| SOC | 1 | −0.882 | 0.267 | 0.250 | 0.912 | 0.1381 |
| TN | 1 | −0.674 | −0.494 | 0.212 | 0.743 | 0.1125 |
| TP | 2 | −0.074 | 0.851 | −0.366 | 0.864 | 0.1309 |
| TK | 3 | 0.413 | 0.428 | 0.669 | 0.802 | 0.1215 |
| Principal component eigenvalue | | 3.510 | 1.716 | 1.376 | | |
| Of Variance (%) | | 43.879 | 21.447 | 17.199 | | |
| Cumulative (%) | | 43.879 | 65.327 | 82.525 | | |

The results showed that deep vertical rotary tillage management could improve the soil quality of saline farmland (Table 6). The scores of eight soil property indices and soil quality indices (SQI) were calculated for each treatment at 0–60 cm of soil for each year (2020–2022) using Equations (4)–(6) and Table 2. Scores under BD, pH, EC, SOC, and TN soil property indexes for each treatment were in the order of CTM < DTM20 < DTM40 < DTM60, with the highest scores obtained under CTM treatment and the lowest scores obtained under DT60 in MWD. The scores were all relatively similar under different tillage management methods in TP. The scores under all treatments were 1.000 because the content of TK was greater than 25 g·kg$^{-1}$ at soil depths of 0–60 cm (Table 2). The magnitude of the soil quality indices (SQI) were in the order of CTM < DTM20 < DTM40 < DTM60, with DTM40 and DTM60 both significantly greater than CTM and DTM20, and DTM60 significantly greater than DTM40 ($p < 0.05$).

**Table 6.** Scores for the soil properties considered in TDS.

| Indicators | Scoring Curve | TDS | | | |
|---|---|---|---|---|---|
| | | Weight | CTM | DTM20 | DTM40 | DTM60 |
| BD | SSF2 | 0.1325 | 0.404 | 0.469 | 0.504 | 0.547 |
| MWD | SSF1 | 0.1035 | 0.652 | 0.644 | 0.618 | 0.594 |
| PH | SSF2 | 0.1254 | 0.353 | 0.364 | 0.384 | 0.409 |
| EC | SSF3 | 0.1356 | 0.272 | 0.305 | 0.549 | 0.675 |
| SOC | SSF1 | 0.1381 | 0.138 | 0.145 | 0.152 | 0.163 |
| TN | SSF1 | 0.1125 | 0.170 | 0.174 | 0.177 | 0.185 |
| TP | SSF1 | 0.1309 | 0.697 | 0.695 | 0.697 | 0.691 |
| TK | SSF1 | 0.1215 | 1.000 | 1.000 | 1.000 | 1.000 |
| SQI | | | 0.453 c | 0.468 c | 0.507 b | 0.532 a * |

* Different letters indicate significant differences at $p < 0.05$.

The results showed that deep vertical rotary tillage management can significantly increase seed cotton yield (Figure 4a). The average seed cotton yield in 2020–2022 was highest under DT60 treatment. There were significant increases of 24.93%, 35.86%, and 43.98% ($p < 0.05$) under DTM20, DTM40, and DTM60, respectively, relative to that of CTM.

The smallest coefficient of variation was 8.53% for CTM treatment, and the highest was 12.65% for DTM60. The Pearson coefficient of SQI and seed cotton yield was 0.93, indicating significant correlation ($p < 0.001$). The $R^2$ value of 0.8658 indicated a good correlation and fit (Figure 4b), indicating this evaluation method is meaningful. Improvements in soil quality can contribute to increased seed cotton yields.

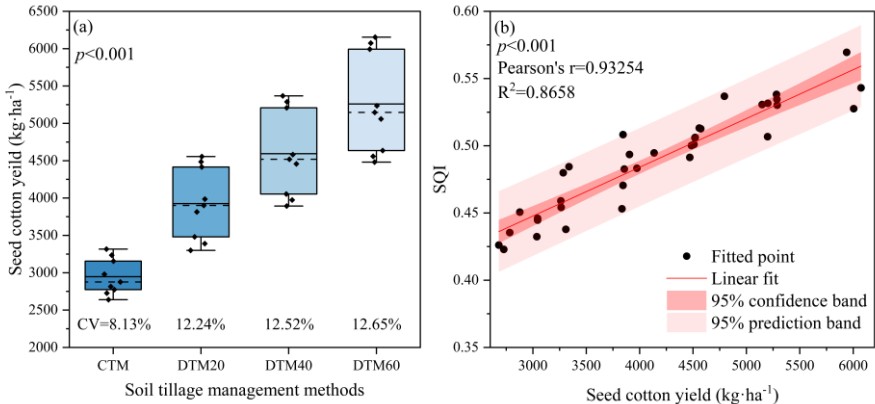

**Figure 4.** Interrelationship of soil quality index and crop yield. *p* is the level of significance of seed cotton yield between treatments in 2020, 2021, and 2022. Figure (**a**) shows seed cotton yield under different tillage management practices in 2020, 2021 and 2022. (**b**) shows the linear fit of seed cotton yield and SQI index.

### 3.4. Relationship between Soil Quality Indicators and Yield of Crops

The relationship between all soil quality indicators and seed cotton yield under different tillage management practices was assessed by principal component analysis (Figure 5). In principal component analysis, PC1, PC2, and PC3 explained 54.60%, 17.80%, and 11.20% of the variance in the data, respectively. BD, pH, and EC were significantly negatively correlated with seed cotton yield, SOC, and TN, indicating that a reduction in BD, MWD, pH, and EC promotes an increase in seed cotton yield, SOC, and TN. None of the relationships between TK and seed cotton yield were significant, and the direction of their vectors was almost perpendicular to the direction of the yield vectors. TP showed a positive correlation with seed cotton yield. Therefore, BD, MWD, pH, and EC were limiting sensitive indicators of seed cotton yield, and SOC and TN were positive sensitive indicators. The data distribution of each treatment shifted leftward toward the vector direction of seed cotton yield with increased depth of tillage. Overall, the results showed that different tillage management practices had significant effects on soil quality indicators and seed cotton yield.

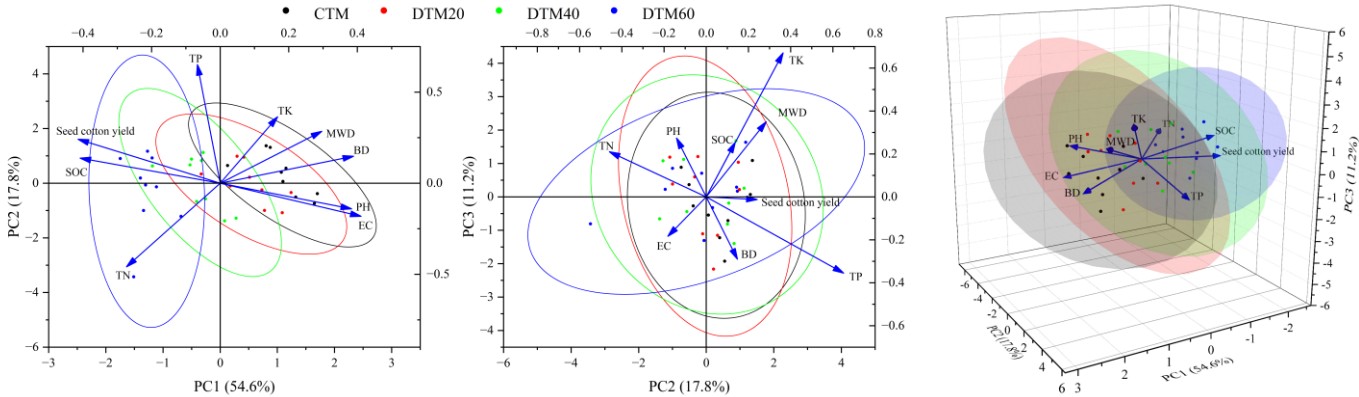

**Figure 5.** Principal component analysis of soil quality indicators and yield of crops in different soil tillage management methods (2D and 3D). The lines originating from the central point of the biplots

show negative or positive correlations of different variables, and their closeness indicates strength of correlation with a particular treatment.

## 4. Discussion

### 4.1. Effect of Different Soil Tillage Management Methods on Soil Quality Indicators

High soil salinity severely limits land productivity and crop production in the southern Xinjiang region [27]. Modified soil tillage management can improve soil properties, and comparing different tillage management helps us to explore ways to improve soil quality in this region. The mean values of BD, MWD, pH, EC, SOC, and TN of 0–60 cm soil were all significantly different under different tillage strategies during 2020–2022, and use of deep vertical rotary tillage sustainably affected the soil structure and environment (Figure 3). Increased depth of deep vertical rotary tillage resulted in greater reductions in BD and MWD, while decreased soil pH and EC resulted in increased levels of SOC and TN. A reduction in soil salinity content by tillage to increase SOC and TN was seen previously [28]. Soil total nitrogen and organic carbon content showed a significant correlation, probably because most of the nitrogen was bound in the organic matter matrix (Figure 3) [29], and the application of organic fertilizer added carbon to the soil. Soil TP content under different tillage management methods did not differ significantly from 2020 to 2022, but TP content decreased in 2020 (Table S1). This was because deep vertical rotary tillage can increase soil permeability and phosphorus leaching, but chemical fertilizer application and straw return to the field can provide carbon, nitrogen, and phosphorus to replenish the nutrients [30]. Different tillage management methods had less effect on TK content, probably because tillage in potassium-rich farmland is always in equilibrium with potassium replenishment and depletion. The results of this study showed that deep vertical rotary tillage decreased the average diameter of soil aggregates (Figure 2). Previous studies have shown that a reduction in soil macroaggregates decreases aggregate stability [31] and also leads to a decrease in carbon and nitrogen contents in the soil [32]. However, in this study, carbon and nitrogen content did not decrease. This may be due to the long-term use of straw returning. The fragmentation of the agglomerates can increase the contact between crushed straw and soil, and this promotes the decomposition of the natural organic matter and enables the soil to obtain more carbon [33–35]. In addition, the crushing of large agglomerates can facilitate the washing out of saline and alkaline ions from the agglomerates under the action of spring irrigation.

### 4.2. Effect of Different Soil Tillage Management Methods on Soil Quality

Soil quality evaluation using sensitivity indicators helps to determine the functional capacity of tillage management in an agroecosystem [36]. There have been few long-term experimental studies using deep vertical rotary tillage. Our results suggest that the effects of deep vertical rotary tillage last longer than three years, as seen by the maintenance of soil structure in the third year. Alternating conventional tillage with deep vertical rotary tillage during this period is a more economical and efficient way to improve saline farmland [37]. Improvement of soil quality can be achieved by enhancing or maintaining soil-related properties, with SOC significantly contributing to soil quality and improving the structural stability and carbon and nitrogen cycling [38]. The results of this study showed that SOC had the highest weight in the evaluation of soil quality, and tillage management practices that scored high in SOC also had the highest SQI. The second highest weight was EC and the lowest was MWD, which is consistent with previous studies [23,39]. Sadiq et al. [40] concluded that deep tillage is important and can reduce soil salinity and promote cotton yield formation. A loose soil structure creates favorable conditions for salt leaching and cotton root growth. The score function for MWD was SSF1, and tillage to a depth that is too deep reduces the average diameter of soil aggregates, explaining the lowest score of the DTM60 under this indicator. The observed scores for total soil potassium under all tillage management practices were all 1.00, indicating that the soil is rich in potassium, sufficient for the soil to reach its full potential. Raiesi and Kabiri [39] found that MWD is less sensitive to tillage than other physical properties, consistent with the results of this study (Table 6).

DTM40 and DTM60 were able to significantly improve soil quality compared to CTM and DTM20, and seed cotton yield showed a significant linear relationship with SQI, indicating the soil quality evaluation was meaningful. Crop yield is directly related to climatic and hydrological factors and field management level, and the soil quality index in this study showed a good fit with seed cotton yield (Figure 5), indicating that improving soil quality is beneficial to crop yield [41].

*4.3. Relationship between Soil Quality Indicators and Seed Cotton Yield under Different Soil Tillage Management Methods*

Soil tillage alters soil organic matter mineralization and nutrient cycling rates [42], which affect soil quality and cotton production. Deep tillage can effectively increase seed cotton yield, and here, seed cotton yield showed a significant linear correlation with SQI. The coefficient of variation (CV) of CTM was the smallest among seed cotton yield, with higher yields and CV values for deep vertical rotary tillage treatments. In the principal component analysis of soil quality indicators and seed cotton yield, BD, MWD, pH, and EC were identified as limiting sensitive indicators for seed cotton yield, and SOC and TN were identified as positive sensitive indicators, consistent with previous results [43]. Therefore, assessing SQI allows determination of the effect of different tillage methods on soil quality. Although soil quality varies in different locations and is associated with different land use and tillage management methods [14], our results clearly suggest that deep tillage management can help to improve saline farmland in the southern Xinjiang region. In the short term, efforts to improve saline soil should focus on desalination, followed by the application of organic fertilizer to improve soil fertility. However, future work should investigate whether the DTM60 tillage management method can result in long-term improvements to saline farmland for crop yield increase. Additionally, future work should investigate the yield increase threshold of this tillage management method. With long-term use, irrigation and fertilization can increase the accumulation of soil salts. Modified tillage management can only temporarily alleviate the limitations of soil salinity on crop yields; the best strategy to solve this problem is to remove the soil salts. Therefore, in future research, we will continue to investigate the effect of deep vertical rotary tillage management and subsurface pipe drainage technology on the improvement of saline farmland. Of course, soil quality is also affected by rainfall and temperature and additional agricultural practices, so future work should investigate the general applicability of the SQI evaluation method.

**5. Conclusions**

Different tillage management methods can significantly affect soil quality indicators and crop yield. The results of this study showed that deep vertical rotary tillage optimizes soil properties and improves the soil quality index (SQI) compared to CTM. Deep tillage reduces BD, MWD, EC, and pH, and promotes the accumulation of SOC and TN in the soil. The average diameter of aggregates in the soil also becomes smaller with an increase in the tillage depth, and a reduction in MWD does not reduce crop yield but promotes soil salinity leaching. The significant linear relationship between seed cotton yield and soil quality ($p < 0.001$) indicated that improving soil quality can increase crop yield. By principal component analysis, we identified BD, MWD, pH, and EC as limiting sensitive indicators for seed cotton yield, and SOC and TN as positive sensitive indicators. Utilizing these sensitivity indicators and evaluating soil quality can help expand our understanding of how to improve land productivity and crop yield potential of saline farmland in southern Xinjiang.

**Supplementary Materials:** The following Supporting Information can be downloaded at: https://www.mdpi.com/article/10.3390/agriculture13101864/s1, Table S1: Soil quality indicators for 0–60 cm, 2020–2022.; Table S2: Seed cotton yield in 2020–2022; Table S3: Data normality distribution test.

**Author Contributions:** Z.L., H.L. and T.W. conceived this study, and H.Y. collected and analyzed the data. All authors contributed critically to the drafts and gave final approval for publication. All authors have read and agreed to the published version of the manuscript.

**Funding:** This research was supported by the National Natural Science Foundation of China (No. 52069026, U1803244).

**Institutional Review Board Statement:** Not applicable.

**Data Availability Statement:** Not applicable.

**Acknowledgments:** Thank you to all participants for their strong support.

**Conflicts of Interest:** The authors declare no conflict of interest.

## Abbreviations

BD, soil bulk density; MWD, soil mean weight diameter; EC electrical conductivity; SOC, soil organic carbon; TN, total nitrogen; TK, total potassium; TP, total phosphorus. CTM, conventional tillage; DTM20, 20 cm; DTM40, 40 cm; DTM60, 60 cm.

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
