# Peer review of "Effects of Deep Vertical Rotary Tillage Management Methods on Soil Quality in Saline Cotton Fields in Southern Xinjiang"

_agriculture, doi:10.3390/agriculture13101864_

Round 1

Reviewer 1 Report

My main comment is that the paper in its current form only addresses the short-term impacts of interventions. The methods tested don't address the fundamental challenges of salinization, where the annual input of salts exceeds the export and salt accumulates. Mixing it through a deeper soil layer will provide temporary relief (and allow higher crop yields), but will lead to a deeper salinized layer of soil in the longer term. Authors could 1) Address this challenge, 2) Try to estimate for how long the deeper tillage might have positive effects (until profile-salinity caught up) and 3) Discuss other ways to tackle the underlying salinity challenge. 

Table 1. Please use standard soil science terms in English, rather than translate Chinese terms. Texture is described as sand, silt, and clay in English.

Author Response

Reviewers' comments

My main comment is that the paper in its current form only addresses the short-term impacts of interventions. The methods tested don't address the fundamental challenges of salinization, where the annual input of salts exceeds the export and salt accumulates. Mixing it through a deeper soil layer will provide temporary relief (and allow higher crop yields), but will lead to a deeper salinized layer of soil in the longer term. Authors could 1) Address this challenge, 2) Try to estimate for how long the deeper tillage might have positive effects (until profile-salinity caught up) and 3) Discuss other ways to tackle the underlying salinity challenge.

Author's Response

First of all, I am very grateful to the reviewers for their suggestions for revising this article. I couldn't agree more with the reviewer's suggestions for this article. Improved tillage management practices can only mitigate salinity damage to the soil surface, but they cannot remove salt from the soil. But as to how long it takes for deep vertical rotary tillage to have a positive impact, I think it may take a longer study to determine that. I have therefore added a related section to the last part of the discussion on other ways to deal with potential salinity challenges.

In my unpublished research, we explored the effectiveness of the combined application of deep vertical rotary tillage and subsurface pipe drainage techniques for the improvement of saline farmland. Spring irrigation with fertility irrigation was able to remove salts from the soil through drainage pipes. This can effectively inhibit the migration of deep soil salts to the surface soil during the non-fertile period, which is an effective way to fundamentally solve soil salinization. However, the cost of this application is high, and further research is needed on more economical ways of applying it.

In addition extensive revisions have been made to address the language problems in the article.

Reviewer 2 Report

The paper is generally well-written and organized, however, some language editing and improvement in word choice, avoiding repetitive sentences and clarifying some statements are needed.  The suggested changes and queries are annotated in the document text attached.  Some specific points for clarification are listed below:

1. On page 2, second paragraph, the term "reasonable tillage management" is used -- this is a bit ambiguous; could you say "improved tillage" or "improved soil and farm management"?

2. Page 3, top paragraph: The repeated sentence should be deleted.

3. In Table 1, Do you mean "silt particles" and "clay particles"? These are the commonly used terms.

4. By "American System of Soil Classification" do you mean USDA Soil Taxonomy system?

5. Of the soil analyses performed, why was "Total P" and "Total K" done instead of "Available P & K"? The latter is usually the common analysis for evaluating crop productivity.

6. Page 4, for determination of soil bulk density, by "ring knife" do you mean a soil corer and core ring was used?

7. Page 6, Results: What is meant by "Tillage age" and "age of tillage"? This is unclear and has not been described fully in the Methods section.

8. Page 7: The results indicate that tillage management did not significantly influence total P and K; this might have been different if "available P and K" were measured.

9. Some long sentences are unclear and could be better expressed in 2 sentences, e.g., Page 10, Discussion section, first sentence.

10. Other language and word modification suggestions are annotated in the document text.

Language and word choice needs improvement as suggested above.

Author Response

Reviewers' comments

The paper is generally well-written and organized, however, some language editing and improvement in word choice, avoiding repetitive sentences and clarifying some statements are needed. The suggested changes and queries are annotated in the document text attached. Some specific points for clarification are listed below:

1.On page 2, second paragraph, the term "reasonable tillage management" is used -- this is a bit ambiguous; could you say "improved tillage" or "improved soil and farm management"?

Author's Response:Revised in accordance with reviewers' comments.

  1. Page 3, top paragraph: The repeated sentence should be deleted.

Author's Response:Revised in accordance with reviewers' comments.

3.In Table 1, Do you mean “silt particles" and “clay particles”? These are the commonly used terms.

Author's Response:Revised in accordance with reviewers' comments. “Powder” has been replaced by "silt" and “Viscous” has been replaced by “clay”.

4.By "American System of Soil Classification" do you mean USDA Soil Taxonomy system?

Author's Response:Revised in accordance with reviewers' comments. Verified that "American System of Soil Classification" has been changed to "USDA Soil Taxonomy system".

5.Of the soil analyses performed, why was "Total P" and "Total K" done instead of "Available P & K"? The latter is usually the common analysis for evaluating crop productivity.

8.Page 7: The results indicate that tillage management did not significantly influence total P and K; this might have been different if "available P and K" were measured.

Author's Response:I am very grateful to the reviewers for their comments, and I will respond uniformly here to reviewers' comments #5 and #8. In the course of our experiment, we found that there was a great spatial and temporal variability in Available P and K under different treatments, which was a problem we had not considered. In addition, we believe that the relatively high levels of phosphorus and potassium in the soil in this experimental area can meet the requirements of cotton growth. We believe that the main reason affecting the conversion of total nutrients to the fast-acting state in the soil is the high salinity content of the soil, which limits this process. In our actual monitoring, the size of the content of fast-acting phosphorus and potassium in the soil was CTM>DTM20>DTM40>DTM60, which was related to the uptake by the cotton root system, which ultimately led to an increase in cotton yield. In our unpublished article, we confirmed this with an increased accumulation of phosphorus and potassium in cotton plants.

6.Page 4, for determination of soil bulk density, by "ring knife" do you mean a soil corer and core ring was used?

Author's Response:Revised in accordance with reviewers' comments. “Ring knife” is a tool for collecting soil bulk density samples. This paragraph has been changed to:Soil samples were collected using cutting-ring method between the wide rows of each treatment to a depth of 60 cm during the cotton harvest in September of each year, and each treatment was replicated three times. Soil bulk density (BD) was calculated using the following formula, which was calculated to obtain the mean value of 0-60 cm:

7.Page 6, Results: What is meant by "Tillage age" and "age of tillage"? This is unclear and has not been described fully in the Methods section.

Author's Response:Revised in accordance with reviewers' comments. The main purpose of this part is to show the effect of tillage age on soil quality indicators, so I changed "Tillage age" and "age of tillage" to "Period of tillage".

9.Some long sentences are unclear and could be better expressed in 2 sentences, e.g., Page 10, Discussion section, first sentence.

Author's Response:Revised in accordance with reviewers' comments.

  1. 9. Other language and word modification suggestions are annotated in the document text.

Author's Response:The reviewers' comments on the linguistic problems in this paper are much appreciated. In addition extensive revisions have been made to address the language problems in the article.

  1. Other language and word modification suggestions are annotated in the document text.

Author's Response:I thank the reviewers for their comments, and I have revised the article for any language problems.

Reviewer 3 Report

In reviewed manuscript, Zhijie Li and co-authors compared consequences of different deep vertical rotary tillage depths set up versus conventional tillage. The design of the experiment is described nice. Result section is too descriptive. I recommend shortening it to avoid repetition with figures and tables. Maybe merging Results and Discussion helps.

My main concern is connected with data processing. I recommend using non-parametric statistics (H-test, Spearman correlation, etc.) instead of parametric (MANOVA, Pearson correlation, etc.). Otherwise, please, provide results of tests for normality of analyzed variables.

L. 27. Please, replace ‘in 0-60 cm of soil’ with ‘in a 0-60 cm layer’.

L. 30 ‘BD’. What is it?

L.101 – 104. Repetition.

L.104 ‘dry ash salt mud’. Please, be more specific and name the soil according to Soil Taxonomy or FAO-WRB.

Table 1. Please, replace ‘Powder ’ with ‘Silt’ and ‘Viscous ’ with ‘Clay’.

L.275 ‘82.525%’. Please, round up here and after in the similar context.

Tables 5 and 6. You do not need to indicate unites for the majority of properties.

L.379 – 380 ‘formation in a The loose’.

L. 386 ‘6.).’

English is well. But some sentences are not easily understandable.

Author Response

Reviewers' comments

  1. The paper is generally well-written and organized, however, some language editing and improvement in word choice, avoiding repetitive sentences and clarifying some statements are needed. The suggested changes and queries are annotated in the document text attached. Some specific points for clarification are listed below:

Author's Response:Many thanks to the reviewers for their comments. The data in this article has been pre-tested for normal distribution. The detailed results have been described in the supplementary material. In addition, in lines 205-211 we have made certain additional explanations.

Table S3. Data Normality Distribution Test

Shapiro–Wilk test

statistic

DF

p-value

BD

0.961

36

0.226

MWD

0.975

36

0.564

PH

0.969

36

0.411

EC

0.944

36

0.067

SOC

0.972

36

0.470

TN

0.950

36

0.105

TP

0.940

36

0.053

TK

0.942

36

0.058

Seed cotton yield

0.959

36

0.198

  1. 27. Please, replace ‘in 0-60 cm of soil’ with ‘in a 0-60 cm layer’.

Author's Response:Revised in accordance with reviewers' comments.

  1. 30 ‘BD’. What is it?

Author's Response:Revised in accordance with reviewers' comments. BD is soil bulk density and has been modified in the abstract. Abbreviated names that will appear below have been specifically labeled in the abstract.

  1. 101 – 104. Repetition.

Author's Response:Revised in accordance with reviewers' comments.

  1. 104 ‘dry ash salt mud’. Please, be more specific and name the soil according to Soil Taxonomy or FAO-WRB.

Author's Response:The naming of the soil types in the area was taken from the Soil Database of China. The naming of the soil name in the original article was translated from Chinese, but there was no English equivalent of the name in the check. The name of this soil type was therefore deleted from the original article. Here's the URL to access the Soil science database: http://vdb3.soil.csdb.cn/extend/jsp/introduction

  1. Table 1. Please, replace ‘Powder ’ with ‘Silt’ and ‘Viscous ’ with ‘Clay’.

Author's Response:Revised in accordance with reviewers' comments.

  1. 275 ‘82.525%’. Please, round up here and after in the similar context.

Author's Response:Revised in accordance with reviewers' comments.

  1. Tables 5 and 6. You do not need to indicate unites for the majority of properties.

Author's Response:Revised in accordance with reviewers' comments.

  1. 379 – 380 ‘formation in a The loose’.

Author's Response:Revised in accordance with reviewers' comments.

  1. 386 ‘6.).’

Author's Response:Revised in accordance with reviewers' comments.

In addition extensive revisions have been made to address the language problems in the article.
